# Conditional Progressive Generative Adversarial Network for satellite image generation

**Renato Cardoso** *
CERN,
1 Esplanade des Particules,
Geneva, Switzerland

Sofia Vallecorsa
CERN,
1 Esplanade des Particules,
Geneva, Switzerland

Edoardo Nemni
United Nations Satellite Centre (UNOSAT),
7 bis, Avenue de la Paix,
CH-1202 Geneva 2, Switzerland

## Abstract

Image generation and image completion are rapidly evolving fields, thanks to machine learning algorithms that are able to realistically replace missing pixels. However, generating large high resolution images, with a large level of details, presents important computational challenges. In this work, we formulate the image generation task as completion of an image where one out of three corners is missing. We then extend this approach to iteratively build larger images with the same level of detail. Our goal is to obtain a scalable methodology to generate high resolution samples typically found in satellite imagery data sets. We introduce a conditional progressive Generative Adversarial Networks (GAN), that generates the missing tile in an image, using as input three initial adjacent tiles encoded in a latent vector by a Wasserstein auto-encoder. We focus on a set of images used by the United Nations Satellite Centre (UNOSAT) to train flood detection tools, and validate the quality of synthetic images in a realistic setup.

## 1  Introduction and related work

The human brain has the amazing ability to detect and complete most missing information using context, e.g. a missing word in a sentence or a missing object in a picture. In a similar fashion, image inpainting or completion [Bertalmío et al., 2000] is the task of replacing pixels in an image, in such a way as to obtain a realistic result, by learning the contextual patterns and structures. The complexity of this task increases with the size and the level of details in a image, since a higher resolution highlights differences between generated and target images [Odena et al., 2016] and it requires, in general, larger computational resources. This is the case with satellite images, which usually cover a wide area with a high amount of details depending on the land topography and the satellite technical specifics.

Our work focuses on high resolution satellite images, typically used by the United Nations Satellite Centre (UNOSAT) for its operations. In spite of the large availability of satellite imagery, tasks requiring high resolution samples suffer from data availability and, often, image sharing limitations due to intellectual property. We propose a image generation tool that frames the task as a iterative completion task, where three quarters of an image are known. In particular, we introduce a conditional progressive growing Generative Adversarial Network that leverages the three corners present in the

---

*renato.cardoso@cern.ch

NeurIPS 2022 Workshop on Synthetic Data for Empowering ML Research.

image to generate the fourth. Once trained this model can be used to iteratively generate larger scenes. The results of this work are preliminary: due to hardware constraints, we train the GAN up to a 256x256 tile. However, results are encouraging and suggest that the methodology can be further developed and extended to produce larger high quality images.

Multiple image generation architectures exist, from variational auto-encoders [Kingma and Welling, 2013] to auto-regressive models [Oord et al., 2016] and Generative Adversarial Networks (GAN) [Goodfellow et al., 2014]. These methods come with different strengths and weaknesses: auto-encoders, for example, are relatively easy to train but tend to produce blurry images; auto-regressive models produce sharp images but are usually slower to evaluate, do not learn the image latent representation (they learn pixel-wise distributions, instead) and, therefore, have a somewhat limited applicability. On the other hand, GANs can produce high resolution images, but tend to learn a rather limited sample diversity while also exhibiting instabilities during training. Nevertheless, GANs remain very popular models and multiple architectures have been developed through the years: style-GAN [Karras et al., 2018] and style2-GAN [Karras et al., 2019], for example, achieve impressive image quality. Previously, [Karras et al., 2017] introduced a progressive growing GAN, addressing the problem of high-resolution image generation, via a training mechanism that focuses on large scale features in low resolution images and gradually adds finer structures as the resolution increases.

The specific problem of image inpainting has also been addressed using different methodologies: for example, the Context Encoder [Pathak et al., 2016], which is one of the first GAN-based inpainting algorithms and the Multi-Scale Neural Patch Synthesis [Yang et al., 2016]. Globally and Locally Consistent Image Completion [Iizuka et al., 2017] is also considered a step up in image inpainting: it introduces the usage of a global and a local discriminator which we adopt in our architecture.

## 2 Conditional Progressive Generative Adversarial Network architecture

Our model, the Conditional Progressive GAN, is inspired by [Karras et al., 2017] and it introduces a conditioning mechanism, based on Wasserstein auto-encoders WAE [Tolstikhin et al., 2017], that encodes the visible tiles in a low dimensional space and feeds them to the generation process. In addition, we introduce a local and a global discriminator (as proposed in Iizuka et al. [2017]) to improve the image coherence. The global discriminator is used to evaluate the image as a whole, while the local discriminator focuses on the inpainted pixels. Together, they contribute to the consistency of the output. The Conditional Progressive GAN architecture is shown in Figure 1.

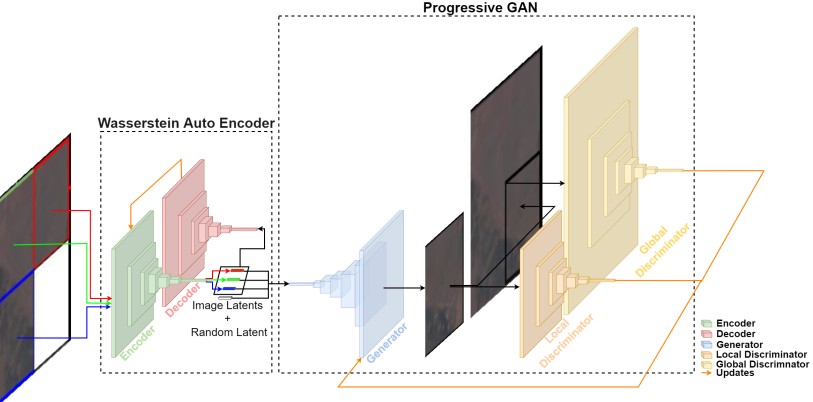

Figure 1: Conditional Progressive GAN architecture including three Wasserstein auto-encoders and a Progressive GAN, with a pair of global and local discriminator.

The Wasserstein auto-encoders also serve as a way to balance the weight of the real tiles latent representations with respect to the random initial seed (both with a 2048 length), while adjusting to the different resolution steps in the progressive training approach. In this test we train four different CNN-based WAEs for the 16x16, 32x32, 64x64 and 128x128 resolution steps. The number of convolutional layers ranges from 3 to 6, while the last layer flattens the convolutional features to

the desired 2048 latent dimension. Batch Normalization and LeakyReLU activation functions are included in the architecture. The Wasserstein auto-encoder uses a WAE-MMD loss [Tolstikhin et al., 2017], defined as the sum of a reconstruction term and a MMD term, using RBF kernels (although we did observe similar results using Inverse Multi Quadratic kernels).

We separately trained each WAE on 2 V100 NVIDIA GPUs for 150 epochs on images belonging to the same data set and down-scaled to the desired resolution. The total training time summed up to about one week with the higher resolution WAE naturally taking up most of the time.

The GAN itself is composed of three elements, a generator and two discriminators as explained above. The generator architecture is inspired by [Karras et al., 2017], with the major difference being the combination of the three input latent vectors with the random seed through a single convolutional layer, at the beginning of the generator network. The local and global discriminators share a similar architecture, with one additional layer in the global discriminator to account for the larger input size. Their outputs are combined into the total training loss through a weighted sum:

$$L_{Combined} = W_{LD} * L_{LD} + W_{GD} * L_{GD} \tag{1}$$

where $L_{LD}$ and $L_{GD}$ correspond to the local and global discriminator losses respectively and $W_{LD}$ and $W_{GD}$ are standard normalization weights defined by

$$W_{LD} = 1 - W_{GD}, \quad W_{GD} \in [0,1] \tag{2}$$

It is important to notice that both discriminator loss terms depend on the quality of the generated tile so they tend to improve simultaneously.

## 3   The data set and the training process

This work uses four Synthetic Aperture Radar (SAR) ESA Sentinel-1 satellite images covering areas affected by floods: they are situated in Myanmar, Cambodia and South Sudan, as shown in tables 3,4 in the appendix. The areas were selected by the United Nations Satellite Centre (UNOSAT) since they capture regions of interest for their operations. ESA Sentinel-1 images are usually quite large with a size of at least 20'000 x 20'000 pixels and a 10-meter pixel resolution. We adopt the same pre-processing procedure defined by UNOSAT in [Nemni et al., 2020]: we split the images in 62474 256x256 tiles and select tiles with at least one flood pixel. This strategy reduces the class imbalance present in the original data, whose distribution is skewed toward background pixels. Two extra images from Nepal are used for testing and validating the network. All images are ortho-rectified and compressed in 8-bit following the steps in [Nemni et al., 2020].

To enable a more robust analysis of the synthetic sample, we train an unsupervised clustering algorithm, Kmeans [Fix and Hodges, 1989], on the original set and obtain four clusters, matching the four common types shown in Figure 3. The first and second types are prevalent and are composed of a mix of small patches of water and dry land. The third is less frequent and it features longer strips of water in different shapes. The last type represents extensive areas of water, typically lakes, and, while it can originate full black tiles, it is much less abundant. The definition of these four clusters is used during the validation step as explained in the next section.

The training process uses 15 million images per step and follows the strategy described in [Karras et al., 2017] starting by generating a 16x16 tile, going up to the desired stopping resolution (128x128 in this test). Again, the training runs on 2 Nvidia V100 cards with 32 GiB memory. The limited GPU memory represents the major constraint to the target resolution in our test, limiting the batch size to 8 for the 128x128 step. Around four weeks are necessary for the full training to complete, with the 128x128 resolution step taking two to three weeks.

Figure 2 shows the local, global discriminator and generator training losses for three resolution values. Given the small batch size for the higher resolution step, the training becomes unstable and, while the overall image quality stays reasonable, as shown further down, the generalisation ability of the model might be affected. A more detailed optimization and training on more powerful hardware is needed.

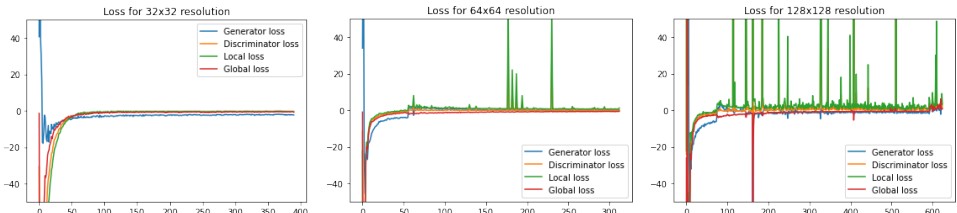

Figure 2: The validation loss for the local and global discriminator and the generator for 32x32 (left), 64x64 (center) and 128x128 (right) generation steps.

# 4   Results and performance validation

We validate the result by first assessing the quality of the generated tiles and then by looking at larger aggregated images.

An initial visual comparison between synthetic and original tiles (figure 3) shows that the GAN output is realistic. We can see that the first three image types are fairly completed: while the first two exhibit higher fidelity, probably due to the larger amount of examples in the data set, the quality of the third one seems lower, although the patch of water is realistically completed. The fourth image type is the least represented in the original data set, thus the patch has a much lower quality. In particular, the edges are clearly visible, suggesting that the relative weight of the global and local losses should be further optimised.

In order to quantify the generator performance we adopt two strategies: one relying on the result of the unsupervised clustering run on the original sample, and the second using the FloodAI Nemni et al. [2020] classifier, currently deployed by UNOSAT for flood detection. As mentioned above, the Kmeans algorithm, trained on the original images, is run on both the test and the generated sample (figure 3); results are, then, compared in order to evaluate whether the GAN is able to realistically reproduce the main image types. In addition, within each of the identified clusters, we calculate the SSIM index as a measure of similarity/diversity for each type. We stress that the quality of the generated images is to be established statistically at the sample level, since we are interested in understanding whether the GAN can be used to generate data that exhibit similar visual features as the original one, while retaining single image originality.  Table 1 collects the SSIM values obtained

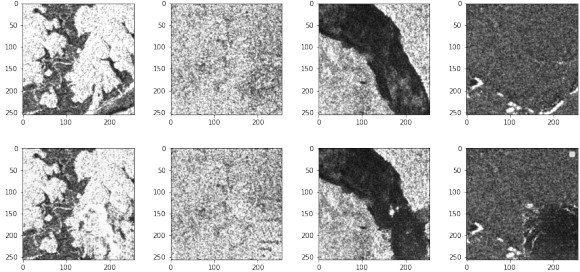

Figure 3: Clusters identified by the Kmeans algorithm. From left to right: small patches of water, patches of dry land, longer water strips and lake-like areas. (top) Clusters in the original data set and (bottom) the synthetic one.

by comparing two images from the original data set (labelled "real/real" SSIM) or two images from the generated data set (labelled "fake/fake"). As expected, the "real/real" sample exhibits the larger diversity, however the SSIM values for the "fake/fake" case are only slightly higher, suggesting that the diversity in the synthetic data is just marginally worse. The "real/fake" case refers to a pair composed of an original image and one from the generated sample. In this case, values are the lowest, hinting at fake images that overall show differences from the original ones (most likely needing better smoothing of the tile edges).

For completeness, and in order to evaluate the robustness of the Kmeans clustering, we compare the outcome of training the algorithm on either real or fake data and find that the results are consistent.

Table 1: SSIM values calculated, as explained in the text, for the four image types identified by the Kmeans algorithm.

|  | Real Data with 4 clusters | | | | Fake Data with 4 clusters | | | |
| --- | --- | --- | --- | --- | --- | --- | --- | --- |
| Real/Real | 0.02 | 0.03 | 0.03 | 0.08 | 0.02 | 0.03 | 0.03 | 0.07 |
| Fake/Fake | 0.04 | 0.04 | 0.05 | 0.11 | 0.03 | 0.05 | 0.05 | 0.10 |
| Real/Fake | 0.01 | 0.02 | 0.02 | 0.05 | 0.01 | 0.02 | 0.02 | 0.04 |
| % images | 29 | 41 | 19 | 11 | 31 | 40 | 18 | 11 |

In addition, we perform the full analysis detailed above configuring the Kmeans maximum number of clusters to three instead of four and, again, we find very similar SSIM values (table 2).

Table 2: SSIM value considering only three Kmeans clusters in the original data set.

|  | Real Data with 3 clusters | | | Fake Data with 3 clusters | | |
| --- | --- | --- | --- | --- | --- | --- |
| Real/Real | 0.03 | 0.02 | 0.06 | 0.03 | 0.02 | 0.06 |
| Fake/Fake | 0.04 | 0.03 | 0.08 | 0.04 | 0.04 | 0.08 |
| Real/Fake | 0.02 | 0.02 | 0.04 | 0.02 | 0.01 | 0.04 |
| % images | 56 | 28 | 16 | 54 | 29 | 17 |

By interpreting the generation step as an iterative completion problem, we can train the GAN on relatively small images, thus saving computing resources, and then use the trained generator to incrementally build a much larger output. Figure 4 shows a 512x512 image obtained by training the conditional progressive GAN up to a tile resolution of 128x128.

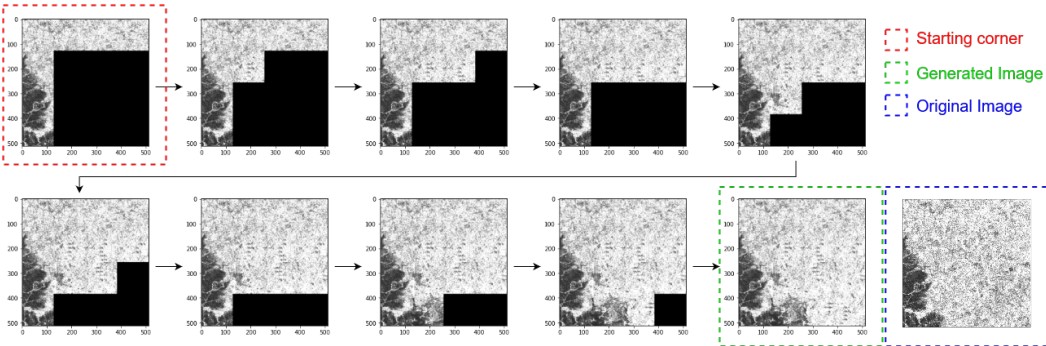

Figure 4: Incremental 512x512 image generation using 128x128 tiles and starting at the top-right most corner.

Additional examples are visible in Figure 5: overall they look realistic, although they seem to hint at increasingly homogeneous new tiles. Additional experiments are required to assess the level of detail retained by each subsequent synthetic tile..

### 4.1 Flood detection through FloodAI

Many image generation tasks evaluate the quality of the output using metrics, such as the Inception Score [Salimans et al., 2016] and the Frechet Inception Distance [Heusel et al., 2017], that rely on independently trained classifiers. However, our model is trained on images exhibiting very specific features and we cannot use generic pre-trained models such as Inception [Szegedy et al., 2014]. Moreover, we are interested in testing the behavior of the synthetic data in a realistic environment and, therefore, we evaluate on it the performance of FloodAI [Nemni et al., 2020], a model producing binary flood/no-flood pixel masks on 256x256 images. Accordingly, we generate 256x256 images

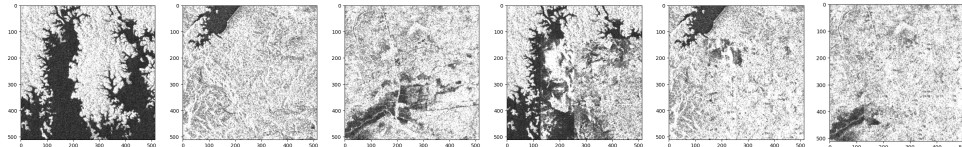

Figure 5: Generation of 512 by 512 images using the iterative process, originals on the left, generated on the right

following the procedure exemplified in figure 4and show the result in figure 6: FloodAI consistently identifies large patches of water as a single element of the flood, as well as identifying small spots of water in the image. This encouraging result suggests that both water and land characteristics are realistically reproduced in the synthetic data set.

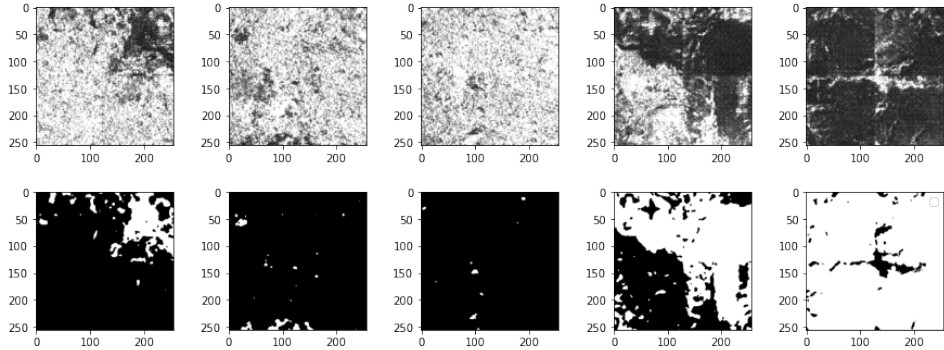

Figure 6: FloodAI test on the synthetic data set. (top) synthetic samples generated by the Conditional progressive GAN. (bottom) FloodAI output mask (water pixels are shown in white).

## 5   Conclusions and Future Plans

The field of generative models develops rapidly and increasingly more complex tasks can be tackled, from text-to-image generation, to hyper-resolution and video generation. This work is intended as a first proof of concept of a methodology for the generation of large high resolution images, such as those typically used in Earth Observation (EO) problems. EO is quickly turning toward data-based techniques, such as machine learning, for an increasing number of tasks and, while public data sets are available through tools such as the ESA Copernicus Hub Portal, use cases remain for which the publicly available data is not sufficient or suitable. At times images resolution is not enough, in other cases the interest lies in studying rare events for which data does not exist. The use of synthetic data sets can also simplify data sharing among different research groups while preventing problems related to restricted use policies, due to either intellectual property or geopolitics. This work proposes a strategy that recasts the image generation process as a iterative completion task. By doing so, it is possible to limit the image size during the training step, thus allowing training on limited hardware, while extending the generation process to much larger sizes without loosing sharpness. With the introduction of a conditional progressive GAN that combines auto-encoders and GAN architectures, we show how to generate a realistic 512x512 image, while training on 128x128 tiles. The result is validated through visual analysis of the single images but also measured at the sample level. In particular we study the composition of the generated sample via a clustering algorithm and we analyse the diversity/similarity of different image types identified both in the generated and the original samples. In addition, and in order to evaluate the quality of the synthetic data set with respect to a realistic, practical application, we run our images through the FloodAI flood detection tool, deployed by UNOSAT. The outcome seems to show that FloodAI processes the synthetic data in the same fashion as the original one. A more conclusive test, would be to retrain FloodAI on the synthetic data and test it on real data, in order to verify whether fake images could be used to successfully reproduce real data in training the classifier. This is one of the next steps in our current research plan. Overall, our results are encouraging, showing a reasonable agreement between the

original and the synthetic data sets in spite of the limited resources used for model development. We plan to continue the development of the conditional progressive GAN, by scaling up the hardware resources and, consequently, the size and resolution of the generated images. In addition we plan to complete a robust assessment of the generalisation properties of the model and diversify the target data sets and tasks.

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

Table 3: Details of the images used in this study.

| Country | No. of pixels |
|---|---|
| Myanmar | 29,409 x 21,601 |
| Myanmar | 30,475 x 27,551 |
| Cambodia | 28,149 x 21,836 |
| South Sudan | 29,055 x 21,812 |
| Nepal | 31,094 x 21,408 |
| Nepal | 31,341 x 21,375 |

Table 4: File names of the images used in this study. The image name can be used directly into the ESA Copernicus Hub Portal `https://scihub.copernicus.eu/dhus/#/home`to download the images.

| Country | Image name |
|---|---|
| Myamar | S1A_IW_GRDH_1SSV_20160805T114607_20160805T114632_012465_0137A7_ACBC |
| Myamar | S1B_IW_GRDH_1SDV_20200802T114539_20200802T114613_022744_02B2A7_1C48 |
| Cambodia | S1B_IW_GRDH_1SDV_20191023T034134_20191023T034159_018597_023095_A182 |
| South Sudan | S1A_IW_GRDH_1SDV_20190923T111155_20190923T111220_029148_034F2E_9682 |
| Nepal | S1A_IW_GRDH_1SDV_20210702T001952_20210702T002017_038591_048DBF_C0E0 |
| Nepal | S1A_IW_GRDH_1SDV_20210707T002758_20210707T002823_038664_049005_98AA |

