# OpenReview forum: "Conditional Progressive Generative Adversarial Network for satellite image generation"
_NeurIPS.cc/2022/Workshop/SyntheticData4ML — Neurips 2022 SyntheticData4ML_

### Official Review · Reviewer_7J29 · 2022-10-08
**Novel approach and important use case. Insufficient experimental results.**

**Rating:** 6
**Confidence:** 5

**Review:**

Summary:
In this paper the authors propose a new approach that conditions a progressive GAN on the latent representations learned by a Wasserstein auto-encoder for image inpainting and then use the proposed architecture to generate a high resolution dataset for flood detection.

Strengths:
The paper is very well written and clear. The target use case of the generated dataset which is flood detection is very important and viable use case and there are lots real applications that can benefit from this dataset. Also the proposed approach can be simply adopted to generate synthetic data for other use cases.
The proposed approach is novel and is clearly explained.

Limitation:
The abstract of the paper suggest that this paper focuses on generating a synthetic image dataset for flood detection. However, there is no experimental results presented to show the detection performance of a flood detection model that is trained on this dataset. The most important way to prove the effectiveness of a synthetic dataset is to show how it improves the performance of the downstream models on the actual target use case.

---

### Official Review · Reviewer_uMCY · 2022-10-15
**The paper tries to iteratively generate synthetic high-resolution image as an image inpainting task in a scalable fashion. It uses adjacent tiles encoded in a latent vector as input for the three corners of the satellite imagery. The architecture incorporates a global and a local discriminator to increase image coherence along with conditioning mechanism motivated by a successful image inpainting approach.**

**Rating:** 5
**Confidence:** 5

**Review:**

Strengths:
- The difficulty increases as the extent of specificity for higher-resolution satellite imagery covering a large area rises. This gives less margin of error between the generated and target image.
- The generator architecture is influenced by [Karras et al., 2017], with the main change being the combining of the three input latent vectors with the random seed at the beginning of the generator network via a single convolutional layer.
- Both the test and the produced sample are subjected to the Kmeans algorithm, which was trained on the original images (figure 3). The results are then compared in order to determine whether the GAN can accurately replicate the specified image categories. The use of statistical measures at sample level by incorporating SSIM index as a measure of similarity/diversity for each type inside each of the detected clusters is done to validate if GAN can be used to generate data that demonstrate comparable visual attributes as the original one while maintaining single image originality.
- This study is meant to serve as a first proof-of-concept for a method for generating large, high-resolution images, like those frequently required for Earth Observation (EO) challenges.
- The author discusses the present importance of limiting the size of the image during the training stage to facilitate training on hardware that is constrained while extending the creation process to much larger sizes without losing sharpness.

Weakness:
- In ‘1. Introduction and related work’, authors discuss the limitations of the proposed results (“The results of this work are preliminary: due to hardware constraints, we train the GAN up to a 256x256 tile”) and speculate on future scope (“However, results are encouraging and suggest that the methodology can be further developed and extended to produce larger high quality images”) with no clear validation. The notion is repeated throughout numerous sections, which is unnecessary.
- The authors obtain comparable results when employing the Wasserstein auto-encoder with MMD term and Inverse Multi Quadratic kernels. However, the later is not picked up without reasoning.
- The research demonstrates a clear concept of unstable training because of the lower batch size for the higher resolution stage. While the overall image quality remains acceptable, the model's generalization ability may suffer. For incomplete ablation studies, the absence of sufficient technology for extensive tuning and training is constantly emphasized. This is also mentioned in the case study of the fourth image type, which is the least represented in the original dataset. The weight of global and local losses that require additional optimization is suggested for the situation.
- The outcome is mostly reported through visual inspection of individual images, but it is also statistically quantified at the sample level. The paper focus on the composition of the created sample using a clustering technique, and examines the diversity/similarity of distinct image categories detected in both the generated and original samples.

Other Comments:
- It would be more appropriate to insert the reference deductive language (lines 106–109) that follows Figure 2 in “Section 5: Conclusion and Future plans”.
- Section 4's opening lines (lines 111–112) require restructuring.
- Correction in the typing error “clusters” in line 140.
- Lacks clarification regarding table 2's relevance and importance.
- Figure 5 can be improved by adding an indentation or border separator for better readability.
- Line 147: Additional experiments are required to assess the level of detail retained by each subsequent synthetic tile.
- Line 161: The sentence appears to be missing a commencing term.
- A study of ablation can be conducted on diverse types of image sources (table 3).
- The literature review is primarily based on archival papers, although it can be improved by including new studies from current publications.

---

### Official Review · Reviewer_nUsP · 2022-10-18
**Needs more improvement but good application domain**

**Rating:** 6
**Confidence:** 3

**Review:**

The authors propose a CGAN-based iterative image completion algorithm for satellite image generation. The results are preliminary training up to 256 x 256

Strengths:

* First proof of concept of a methodology for the generation of large high-resolution images, such as those typically used in Earth Observation (EO) problems


Weaknesses:

* No ablation study
* Oscillations
* Even though SSIMs are close in Tables 1 & 2, diversity in the synthetic data is worse

---

### Meta-Review · Area_Chair_mcmg · 2022-10-19

**Recommendation:** Accept